# Navigating Work Career through Locus of Control and Job Satisfaction: The Mediation Role of Work Values Ethic

**Claire A. Simmers [1],* and Adela J. McMurray [2]**

[1]   Department of Management, Saint Joseph's University, Philadelphia, PA 19131, USA
[2]   College of Business, Government and Law, Flinders University, Bedford Park SA 5042, Australia
*   Correspondence: simmers@sju.edu

**Abstract:** This study examines navigating work careers through self-concept (locus of control and work values ethic) and job satisfaction within the postindustrial work environment of the 21st century. Career construction theory conceptualizes one's career as a process of responding to a changing environment through self-concepts to more actively construct their careers. The SARS-CoV-2 (COVID-19) pandemic further highlights the importance of individuals' self-leading their work journeys. The study indicates that work values ethic is an important variable in further explaining the relationship between locus of control and job satisfaction in our sample. When the effect of work values ethic is removed, the association between locus of control and job satisfaction is insignificant. We expect this research to spur further efforts by individuals to improve their understanding of the intricacies among their intrapersonal traits, needs, and abilities to better navigate their work careers with application to newly defined workplaces as a result of COVID-19. Our results also inform the practice of career education and counseling.

**Keywords:** work careers; locus of control; work values ethic; job satisfaction; career construction theory



## 1. Introduction

While individuals have been responsible for their well-being in the workplace, the technological boom of the 21st century, globalization, and job redesign have accelerated the rate and depth of the challenges facing individuals in navigating their work careers [1]. Full-time jobs with one employer during all or a substantial part of one's career are not the norm [2,3]. There is an increase in job switching (churn) either voluntarily or involuntarily [4,5], and there are an increasing number of individuals working primarily as short-term independent freelance workers [6]. Layered onto these work career vicissitudes is the undeniably unprecedented pandemic of SARS-CoV-2 (COVID-19) that burst upon the world in 2020 and continues to impact how we live and work [7].

In this workplace of churn, independent work, and a pandemic, career choice and development are even more of a continual process, with cycles of adjustments linked either to an individual's dissatisfaction with a job [8] or organizational decisions such as downsizing, layoffs and cost-cutting [9], or external factors [10]. These new conditions of work life highlight that a career belongs to the person, not the organization [11]. Whether individuals are either job-hopping to escape from a negative situation, to accelerate their career, or because they were laid-off or find themselves part of the independent workforce, few explore the complexities of interactions among their intrapersonal traits, needs, and abilities [2,9] to proactively build and navigate work careers. There is substantial opportunity for researchers to contribute to our increased understanding of job satisfaction as a motivator in navigating work careers [6,11]. Prior research on job satisfaction partially recognized the role of individual personality and attitudes in the workplace [12,13]. However, the responsibilities of organizations for working conditions; organizational policy, strategies, and promotion; job stress; and compensation packages are more researched [14,15].

Understudied is the topic of job satisfaction, where an increasing number of workers are either voluntarily or involuntarily job-hoppers, are independent workers, or face changes in work due to the pandemic. This may be due, in part, to accessing and researching this work population, but it may also arise from confusions by many about the current nature of the work accomplished and their career choices [6,7].

The context for many career development theories in the 20th century was primarily anchored in organizational employment situations [9]. While the self-concept theory of career development suggested that career choice was a process of self-concept development and implementation [16], the theory of career construction development extended self-concept theory for use within the current 21st century work context [11]. In expressing vocational preferences, individuals conceptualize occupational choice as implementing self-concept. Individuals see work as an indicator of self-worth, and vocational progress as a better integration of person and context. Work is a setting for human development and is a critical component in our adult life. Even if the work choices are imperfect, many identify chances to find some constructive experiences [11,17].

Job satisfaction is a "pleasurable or positive emotional state resulting from the appraisal of one's job or job experiences" ([18], p. 1304). It is a key element of work well-being and is related to work behaviors such as organizational citizenship [19] and job performance [20]. Job satisfaction can also mitigate work withdrawal actions such as absenteeism [21] and turnover [22]. Studies have shown that job satisfaction can also somewhat mediate the relationship of unconstructive personality traits and aberrant work activities [23]. Job satisfaction also serves as a connection between supervisory behavior and organizational commitment [24]. Job satisfaction is significant in career development; it rises when there is an alignment between personal career plans and workplace career fit [25].

A limitation of prior research addressing job satisfaction is the lack of a better understanding of antecedent self-concepts which contribute to job satisfaction within the changing work environment of the 21st century. These relationships are important to understand, as the individual is increasingly more responsible for their job satisfaction; the more self-aware an individual is about their aptitudes, personal style, and work ethics, the greater the potential to enhance or seek job satisfaction as part of their career development. Locus of control is a measure of self-concept where people evaluate whether there is a direct connection between their actions and what happens to them. Those who see a closer linkage have more internal locus (they think they control outcomes) and those who see less of a linkage have an external locus (they think that forces outside of their person control outcomes) [26]. It is possible that those who think they control outcomes (internal locus of control) will experience higher levels of job satisfaction, regardless of their career situation. Work values ethic is a dispositional self-attribution of work success to individual effort. Individuals expressing a strong work values ethic ascribe positive work outcomes to their exertions; they believe that work success primarily rests with individual efforts. As such, they often tolerate their current work situation, as they see it as building career experiences, even if it is not their preferred career place [27].

The present study seeks to address a gap in understanding job satisfaction in our contemporary work environments characterized by 24 × 7 connectivity, globalization, importance of the freelance economy, job insecurity, and continual flux [28]. In this work atmosphere, individuals are increasingly responsible for directing their own well-being. This study applies the causal agency theory from the special education and positive psychology literatures [29], career construction theory [11], and the dispositional theory of job satisfaction [30] to answer the research question of whether locus of control has either a direct influence on job satisfaction or if there is an indirect relationship through work values ethic. Figure 1 shows the research model for this study. In the next sections, the concepts and rationale for their place in the model are explained.

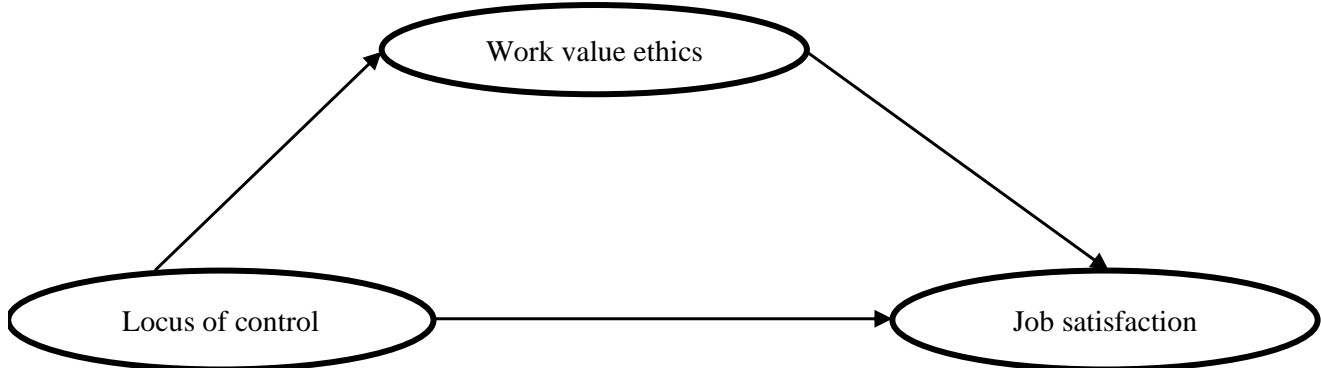

**Figure 1.** Research model: locus of control, work values ethic, and job satisfaction.

*1.1. Indicators of Self-Concept*

1.1.1. Locus of Control

Locus of control is an individual difference construct focusing on beliefs about the degree of control people exert over what happens to them. It has been widely studied for more than half a century, and while locus of control was considered important in work-related relationships, some questioned the role of locus of control as an independent predictor. However, recent research advocates that locus of control is an independent concept and that researchers should continue including locus of control as a discrete construct [31].

Locus of control is a concept related to determinations of why outcomes of events occur and is based on personal insight about the source of outcomes in their lives [26]. Locus of control refers to the degree to which individuals have confidence in their abilities to control the consequences of events and in the ascription of failed outcomes [32]. Those who see external forces as directing actions are identified as having an external locus of control; they primarily think that luck, chance, or other people control their outcomes. Those with an internal locus of control are more likely to have accountability for their behaviors. They work hard to achieve their goals and are self-reliant when challenges arise. They see their efforts as primary causes of their successes and/or failures [33]. Previous research indicates that those with an internal locus of control have higher job satisfaction, more respect for their leaders, feel less pressure in their roles, are more self-directed, and stay longer with their organizations than those with an external locus of control [34]. Those with an internal locus of control reported that an investment in their development would produce a higher pay-off [35], and they were more likely to make long-term investments in personal health and to be entrepreneurs [36] than those with an external locus of control. Additionally, those with an internal locus of control had higher performance levels in work and personal lives, were more likely to seek advanced education [37], and reported better relationships with their supervisors [38] than those with an external locus of control. Attitudes toward unemployment also varied between those with an internal locus versus those with an external locus of control. Those with an internal locus of control were confident that their job search efforts would generate numerous job offers, and they secured jobs with higher wages than those with an external locus [39]. Overall, higher levels of job satisfaction have been linked to those reporting having an internal locus of control rather than those reporting having an external locus of control [40].

1.1.2. Work Values Ethic

Work values ethic defines the importance of individual hard work and frugality in achieving positive psychological, religious, and economic outcomes. Work values ethic seeks to trace the attribution of success and failure in the workplace to individual efforts, and it is a general aspect of work commitment with a focus on attributions to individual effort in the work realm to achieve desired positive work outcomes [41]. It provides

individuals the framework for decision making regarding important goals [42]. Previous research identified explanatory antecedents to work values ethic (GDP per capita, birth nation, and immigrant status) and indicated that work efforts could gain positive outcomes such as power, prestige, and wealth [27]. These findings were consistent with prior work where individuals from developing nations tended toward higher work values ethic scores than those from developed nations [43]. Additionally, it has been reported that those with a strong work values ethic often do not require support from superiors to attain career and job satisfaction [44]. The work values of individualism and willingness to take risks were related to various aspects of job satisfaction [45]. An individual's values are important ways to cope with and better understand the rapidly changing work contexts and are vital to navigating careers [11].

### 1.2. Job Satisfaction—Indicator of Well-Being

Job satisfaction is a very popular topic, studied by researchers in many disciplines. It represents the affective reactions of individuals toward their work and work life and represents a constructive frame of mind concerning a job [46]. Job satisfaction can affect employment situations and influence organizational productivity, employee absenteeism, and turnover [47], and is a strong influence on overall well-being [48].

There are several main explanatory theories used to examine the phenomenon of job satisfaction. One of the earliest is equity theory, which posits that individuals seek fairness in relationships, including those within a work setting. Employees will be satisfied with their jobs if they perceive equity in interactions among themselves, their employer, and other employees [49]. Range of affect theory has some support and states that job satisfaction is determined by a discrepancy between what one wants in a work situation and what one has in that work situation [50]. The two-factor theory (also known as motivator-hygiene theory) attempts to explain satisfaction and motivation in the workplace by stating that satisfaction and dissatisfaction derive from motivation and hygiene factors [51]. The job characteristics model theorizes how five organizationally based job characteristics (skill variety, task identity, task significance, autonomy, and feedback) influence a variety of work outcomes, including job satisfaction [52]. These theories have limited applicability to this study since they generally focus on organizational contributions to job satisfaction. Lastly, the dispositional model of job satisfactions posits that job satisfaction is a personal attitude and will vary from person to person; as a fundamental attitude, it is generally stable across an individual's careers and time [53]. Employers seek to foster job satisfaction among their employees, as satisfied workers are more likely to have lower attrition and higher productivity [46]. However, the dispositional model posits that organizational support is necessary but is insufficient, as individual affective states have a stronger effect on job satisfaction. In the 21st century workplace, the responsibility for job satisfaction may increasingly be with the individual, as the freelance economy and job instability become more the norm than the exception. Thus, an individual's self-evaluation and work ethics may be important influencers of job satisfaction [54]. The following sections detail how this relationship is conceptualized, the methodology we used to test these relationships, our results, discussion, and conclusion.

### 1.3. The Relationship between Locus of Control, Job Satisfaction, and Work Values Ethic

We examined the relationship between the self-concepts of locus of control and work values ethic and well-being (job satisfaction) within the context of a changing work environment [55]. We applied causal agency theory from special education and positive psychology [29] and the dispositional theory on job satisfaction [30] to describe and justify the model relationships. Causal agency theory builds upon the functional model of self-determination to explain how individuals who learn by reacting in some situations gain confidence to proactively be causal agents in other situations. This cycle of reacting and acting then affects perceptions of successfully addressing basic needs, and general

well-being is enhanced. Individuals gain increasing personal empowerment as they achieve their goals. There are three types of action-control belief:

(1) Beliefs about the connection between the self and the goal (control expectancy: 'When I want to do something, I can');

(2) Beliefs about the association between the self and the means for achieving the goal (capacity beliefs: 'I have the capabilities to do something'); and

(3) Beliefs about the utility of a given means for achieving a goal (causality beliefs: 'I believe my effort will lead to goal achievement' vs. 'I believe other factors will lead to goal achievement') ([29], p. 259).

### 1.3.1. Locus of Control and Job Satisfaction

The dispositional theory model posits that job satisfaction is an individual trait that is, to some extent, stable; the genetic base is refined throughout a lifetime and a variety of career and job experiences [53]. Previous research indicated that self-concepts such as an internal locus of control [30] and a Protestant Work Ethic and Confucian values [56] contribute toward higher job satisfaction. A meta-analysis study suggested that self-concepts are among the best dispositional predictors of job satisfaction and job performance [57]. Another study found that the core self-evaluation (a component of internal locus of control) of supervisors and followers positively related to job satisfaction [32]. Results of several other studies [33,58,59] indicated that people with an internal locus of control expressed higher job satisfaction than those with an external locus of control. In addition, various scholars identified the direct relationship between locus of control and job satisfaction. For example, a study in Taiwan [60] found that locus of control plays an incremental role in influencing accountants' job satisfaction, performance, and stress at CPA organizations. In Bangalore [61], a study reported a positive relationship between internal locus of control and job satisfaction. External locus of control was linked to low job satisfaction among middle managers belonging to various sectors (semigovernment level) in Oman [62]. We identify locus of control with causality beliefs from the casual agency model and dispositional theory and hypothesize the following:

**Hypothesis 1.** *Internal locus of control is positively associated with higher levels of job satisfaction.*

### 1.3.2. Locus of Control and Work Values Ethic

This influence may not be a simple direct relationship as posited in Hypothesis 1. We were interested in investigating why internal locus of control possibly has a positive association with job satisfaction. A possible pathway through which locus of control may exercise constructive effects on psychological well-being is belief in the importance of work values ethic. Theoretical arguments suggest that work values and/or their correlates often assume a mediating role, intervening between the effects of background personality variables and affective states, and it is possible that internal locus of control will be linked to stronger beliefs in the importance of work through control expectancy [29].

### 1.3.3. Work Values Ethic and Job Satisfaction

If work values ethic is a pathway through which locus of control has positive influence on job satisfaction, we sought to test if work values ethic is directly related to job satisfaction. Specifically, the benefits of hard work range from job success [63] to the importance of work in its own right [64,65]. Additionally, compared with unemployed individuals, those who are employed displayed better psychological health [65]. Further work values ethic facilitates individuals to achieve career and job satisfaction because such individuals rarely rely on their supervisors to achieve job satisfaction [44]. We hypothesized the following:

**Hypothesis 2a.** *The relationship between internal locus of control and job satisfaction is mediated by work values ethic.*

**Hypothesis 2b.** *There is a positive relationship between work values ethic and job satisfaction.*

## 2. Materials and Methods

### 2.1. Sample

The data were obtained by an anonymous survey using Amazon Mechanical Turk (AMT). AMT has become an acceptable method in academic research [66,67]. There were 282 (97%) usable surveys in the sample. As a further screening, inclusion in the sample depended upon participants choosing the correct answer to three attention check questions placed throughout the survey. A high percentage of the respondents had a bachelor's, a master's, or a professional degree, and under half worked for companies with more than 250 workers. A quarter of the participants worked in the financial/insurance/real estate area, with twenty percent each in trade and education A third of the respondents worked in professional positions, while an equal number reported their positions as middle-level managers or lower-level managers (15% each), and a tenth indicated they worked in technical positions. More than half of the participants had 5 years or less in their present positions, and a third reported 5–10 years. Thirty-nine percent reported 1–5 years employment in their current organizations, and thirty-eight percent reported 5–10 years. The participants were overwhelmingly born (93%) and working in North America (98%). Two-thirds of the participants were born between 1980 and 1999, and females comprised 45% of the sample. The sample had sufficient variation across the demographic information, except for birth country and current workplace, which were not unexpected since English was required as the primary language and only those in North America had access to the survey. Analysis of variance tests revealed no differences in job satisfaction scores among the various demographic categories, so these variables were not needed as controls in subsequent analyses.

### 2.2. Measures

Measures in the study were drawn from previous research. The Rotter instrument [26] was used to measure locus of control. Each of the 29 items comprised a pair of alternatives lettered "a" or "b". Respondents were asked to select one statement from each pair (and only one) which they believed to be most true for them. Every "a" item response was coded as a "1"; the "a" statement represented external locus of control. Scores ranged along a continuum of 0 (very strong internal locus of control) to 29 (very strong external locus of control).

Work values ethic was assessed by the five-item scale to measure the extent that work success was attributed to individual effort [27]. A modified five-item scale was used to assess job satisfaction [30]. Both measures had high average internal consistency reliability, 0.84 for work values ethic and 0.91 for job satisfaction, and thus were considered reliable [68].

### 2.3. Analyses

To test our hypotheses, we analyzed the data with a mediation model using structural equation modeling (SEM). The advantage of using SEM is that one can test the existence of direct effects and indirect effects, which we predicted in our research model (Figure 1).

## 3. Results

Table 1 presents the descriptive analysis of locus of control, work values ethic, and job satisfaction. Table 2 shows the items and the results of confirmatory factor analyses. The items load onto a single factor for work values ethic and a single factor for job satisfaction. There is good reliability and high internal consistency, as each item is above 0.5 and the comparative fit index (CFI) values are 0.97, indicating a well-fitting model [69].

**Table 1.** Descriptive analysis of locus of control, work values ethics and job satisfaction.

| Variable | Observations | Mean | Standard Deviation | Min | Max |
|---|---|---|---|---|---|
| Locus of Control | 282 | 10.482 | 4.853 | 0 | 21 |
| Work Values Ethic | 282 | 3.374 | 0.474 | 2.2 | 5 |
| Job Satisfaction | 282 | 3.640 | 0.975 | 1 | 5 |

**Table 2.** Factor loading for work values ethic and job satisfaction.

| Variable | Factor1 | Uniqueness |
|---|---|---|
| Work Values Ethic | | |
| Most people who don't succeed in life are just plain lazy | 0.732 | 0.372 |
| A distaste for hard work usually reflects a weakness of character | 0.611 | 0.621 |
| Any person who is able and willing to work hard has a good chance of succeeding | 0.733 | 0.341 |
| People who fail at a job have usually not tried hard enough | 0.746 | 0.349 |
| If one works hard, one is likely to make a good life for oneself | 0.778 | 0.297 |
| Job Satisfaction | | |
| I feel fairly satisfied with my present job. | 0.859 | 0.243 |
| Most days I am enthusiastic about my work. | 0.888 | 0.182 |
| Each day of work seems like it will never end. | 0.682 | 0.430 |
| I find real enjoyment in my work. | 0.886 | 0.181 |
| I consider my job rather unpleasant. | 0.789 | 0.303 |

Table 3 shows evidence that a higher score on locus of control (more external LOC) will result in higher level of work values ethic. The relationship between locus of control and work values ethic is positive and significant ($\beta = 0.088$, $p < 0.01$). The results are robust by controlling for correlations between the residuals of dimensions of work values ethic. The goodness-of-fit statistics of the model are satisfied with the CFI of 0.992 and the Tucker–Lewis index (TLI) of 0.982. These values are in line with the values suggested by earlier scholars [70].

**Table 3.** Locus of control and work values ethic.

| Variables | (1) Work Values Ethic |
|---|---|
| Locus of control | 0.088 ** |
| | (0.011) |

Note: ** $p < 0.01$, standard errors in parentheses; estimation of other factors is not shown.

Table 4 shows that the relationship between work values ethic and job satisfaction was negative and significant ($\beta = -0.445$, $p < 0.01$), with solid results of goodness-of-fit statistics (CFI = 0.981, TLI = 0.974). Thus, hypothesis 2b is partially supported, as there is a significant relationship, but not in the predicted direction. The relationship between locus of control and job satisfaction is not significant; hypothesis 1 is not supported.

**Table 4.** Locus of control, work values ethic and job satisfaction.

| Variables | (1) Job Satisfaction |
|---|---|
| Work values ethic | −0.445 ** |
| | (0.128) |
| Locus of control | −0.027 |
| | (0.015) |

Note: ** $p < 0.01$; standard errors in parentheses; estimation of other factors is not shown.

The direct and indirect impacts of locus of control on job satisfaction are examined in Table 5. It can be seen that the indirect impact ($\beta = -0.0376$, $p < 0.01$) of locus of control on job satisfaction through work values ethic is stronger than the direct impact ($\beta = -0.0272$, $p$ = ns). The direct effect of locus of control becomes insignificant with work values ethic as a mediating variable. Therefore, the impact of locus of control on job satisfaction is through the mediation effect of work values ethic. The indirect effect of locus of control through work values ethic was negative and significant ($\beta = -0.0376$, $p < 0.01$). Hence, complete mediation exists. The relationship between locus of control and job satisfaction was mediated by work values ethic, thus supporting Hypothesis 2a.

**Table 5.** Direct effect and indirect effect-locus of control, work values ethic on job satisfaction.

| Locus of Control | Job Satisfaction |
|---|---|
| Direct effect | −0.0272 |
| | (0.0149) |
| Indirect effect | −0.0376 ** |
| | (0.0106) |
| Total effect | −0.0648 ** |
| | (0.0117) |

Note: ** $p < 0.01$ standard errors in parentheses; estimation of other factors is not shown.

## 4. Discussion

### 4.1. Interpretation

Career construction theory framed our examination of the impact of self-concept (locus of control and work values ethic) on a measure of well-being (job satisfaction) within the changing work environment of the 21st century. These relationships are important to understand as the individual increasingly becomes more responsible for their own well-being. The more self-aware an individual is about their aptitudes, personal style, and work ethics, the greater the potential to manage job satisfaction. In the current work atmosphere, the management of personal and work relationships and issues is more complicated because of the overlap of work and personal and challenging organizational and external forces. Our study contributes to a growing understanding of how we manage careers in this dynamic environment.

The results indicate support for Hypothesis 2a; the importance of the self-concept of locus of control on job satisfaction is mediated through work values ethic. The self-evaluation of being in control and perceiving that job satisfaction is under ones control is achieved in our sample through valuing individual hard work and frugality, and is consistent with both the causal agency theory [29] and the dispositional theory of job satisfaction [30]. The results do not support a direct relationship between internal locus of control and higher levels of job satisfaction as found in previous research [34,59,60].

Differing from our review of the literature and relevant theories, the relationship between locus of control and work values ethic was positive, indicating that those with a more externally focused locus of control held the attribution that individual hard work was important. Theory would suggest the opposite, that those with an internal locus of control are more apt to accept accountability for their own behaviors and would be related to stronger levels of work values ethic. One possible explanation is that locus of control is a generalized self-concept about life, while work values ethic could be specific to a workplace. Respondents that reported trending toward an external locus of control in many aspects of their lives could believe that in their specific workplace, work success could still be attributable to their efforts. Additional work should be conducted to investigate this possibility of general self-concepts versus specific workplace success attributions.

A second surprising result is the negative relationship between work values ethic and job satisfaction. This finding suggests that there might be a dissonance between strong attributions of work effort affecting success and an appraisal of one's actual job satisfaction. One could believe that work values ethic is critical to success, but that individual efforts

are not being rewarded or recognized in the current work situation [71]. Additionally, emphasizing work efforts could negatively impact other aspects of the job situation such as social relationships, creativity, and leisure [72], creating incompatible pressures that cannot be resolved by working harder. This is an area where career building can be viewed as a process with the help of career guidance professionals. If the current work of an individual may not be ideal, it can still contribute to building life themes and glean meaning from each work experience [11]. This may be particularly relevant to those working either full-time or part-time as independent workers.

The study results show that work values ethic is an important variable in that it explains why there is a relation between locus of control and job satisfaction in our sample. When the effect of work values ethic is removed, the relation between locus of control and job satisfaction is statistically insignificant ($p > 0.10$) [73]. Collectively, the results further inform the complexity of career construction theory by demonstrating the intricate relationships between locus of control, work values ethic, and job satisfaction in today's tenuous workplaces.

### 4.2. Limitations and Future Research Directions

Further research is needed to examine the generalizability of the results to a wider population than the sample in this study. Such efforts should also investigate well-being and performance outcomes among those employed full- or part-time in the freelance economy and examine those affected by COVID-19 workplace changes such as virtual work, redesigned work, and increasing work–life balance challenges. Further work should also extend into cross-cultural aspects, especially with increasingly global work force mobility and digitally enhanced work capabilities fostering global work teams. Additional research could counter a limitation in this study by including important missing variables such as additional self-evaluations measures, mediational variables, well-being measures, and performance outcomes. A mixed-methods approach would also counter the quantitative, cross-sectional nature of this study.

### 5. Conclusions

Through the lens of career construction theory, this study extends prior research on self-concept and well-being in several ways. It illustrates the importance of work values ethic in influencing the nature and consequences of the locus of control–job satisfaction dynamic in the changing and ever-more-challenging environment of current work. The findings indicate that individuals may value individual hard work and frugality, but it might not be enough to counter other individual and organizational role pressures and constraints encountered. This needs additional investigation, especially among those employed in the freelance economy [74] or facing upheavals during the pandemic [75].

We expect that our study will stimulate those researchers and practitioners in identifying the ways in which individuals navigate their careers in the changing work context. This is particularly relevant in navigating a COVID-19-impacted workplace [76]. Career educators and counselors may benefit from a deeper understanding of the determinants and consequences of well-being in different types of employment situations. Refocusing the goals of career interventions is important in a workplace of increasing diversity in categories of workers who are organizationally employed but frustrated or advancing, and those employed in the independent sector.

**Author Contributions:** Writing—original draft and final draft, C.A.S.; Writing—review & editing, A.J.M. All authors have read and agreed to the published version of the manuscript.

**Funding:** This research received no external funding.

**Informed Consent Statement:** Informed consent was obtained from all subjects involved in the study.

**Data Availability Statement:** The data that support the findings of this study are available on request from the corresponding author.

**Conflicts of Interest:** The authors declare no conflict of interest.

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
