# Peer review of "Navigating Work Career through Locus of Control and Job Satisfaction: The Mediation Role of Work Values Ethic"

_merits, doi:10.3390/merits2040018_

Round 1

Reviewer 1 Report

the study - 1914570 - is of interest to researchers and was conducted appropriately. The results are interesting and well-supported. As the author(s) point out, the surprising result about work values ethic and job satisfaction should lead to future research, perhaps qualitative studies, to understand the 'how' and 'why' of the role of work value ethics to job satisfaction. 

The manuscript is a good example of conducting a study and writing the results in a manner that sees the value of unexpected results.

the study was well-done and the manuscript well-written. 

I don't have recommendations on making the manuscript better. Any suggestions would just be personal preference.

Author Response

Thank you for your kind comments and your time in reviewing.  

Reviewer 2 Report

The paper presents an empirical study on the effects of the relationship between self-concept (locus of control and work values ethic) and job satisfaction on the development of the professional career within the postindustrial work environment of the 21st century.

Although the purpose of the study is adequately expressed, the paper does not clearly mention the main question that this study addressed. Would be indicated to introduce a part or a phrase, before the part named ”The relationship between locus of control, job satisfaction and work values ethic”,  which will state what is the main question (e.g. if the locus of control, as a self-concept indicator can directly predict job satisfaction, regardless of other variables related to self-concept) and what is the specific gap in the field addressed in the study.

Overall, the study is relevant for the field and presented well-structured; the research design is appropriate to test the hypothesis and the data are properly shown. The conclusions are consistent with the evidence and arguments presented. Although the research methodology is sufficiently detailed, does not offer any information about measures of research ethics involved in the study.  Also, the cited references are relevant for the idea expressed, but they are not the most recent publications (within the last 5 years).

Author Response

Thank you for your helpful comments.

  1. We added the gap and research question as you suggested on page 3: 

    The present study seeks to address a gap our understanding of job satisfaction in our contemporary work environments characterized by 24x7 connectivity, globalization, importance of the freelance economy, job insecurity, and continual flux [28]. In this work atmosphere, individuals are increasingly responsible to direct their own well-being.  This study applies the causal agency theory from the special education and positive psychology literatures [29], career construction theory [11] and the dis-positional theory of job satisfaction [30] to answer the research question if locus of control has either a direct influence on job satisfaction or if there is an indirect relationship through work values ethic.

  2. We also added ethics statements in the Supplementary Materials section at the end of the paper: 

    Institutional Review Board Statement: Ethical review and approval are waived due to the complete anonymity of the study.

    Informed Consent Statement: Informed consent was obtained from all subjects involved in the study.

    Data Availability Statement: The data that supports the findings of this study are available on request from the corresponding author.

    Conflicts of Interest: The authors declare no conflict of interest.

  3. We reviewed the references and deleted older ones if a newer one conveyed updated information.  However, we did include some of the original works especially for job satisfaction, locus of control and career theory since these works are important to show the depth of the research and how this study builds on previous research to address current workplace challenges.